# Giant Cell Arteritis: Advances in Understanding Pathogenesis and Implications for Clinical Practice

**DOI:** 10.3390/cells13030267

**Published:** 2024-01-31

**Authors:** Marino Paroli, Rosalba Caccavale, Daniele Accapezzato

**Affiliations:** Department of Clinical, Internal, Anesthesiologic and Cardiovascular Sciences, Sapienza University of Rome, Polo Pontino, 04100 Latina, Italy; rosalba_caccavale@yahoo.it (R.C.); daniele.accapezzato@uniroma1.it (D.A.)

**Keywords:** giant cell arteritis, innate immunity, adaptive immunity, cytokines, vascular wall cells

## Abstract

Giant cell arteritis (GCA) is a noninfectious granulomatous vasculitis of unknown etiology affecting individuals older than 50 years. Two forms of GCA have been identified: a cranial form involving the medium-caliber temporal artery causing temporal arteritis (TA) and an extracranial form involving the large vessels, mainly the thoracic aorta and its branches. GCA generally affects individuals with a genetic predisposition, but several epigenetic (micro)environmental factors are often critical for the onset of this vasculitis. A key role in the pathogenesis of GCA is played by cells of both the innate and adaptive immune systems, which contribute to the formation of granulomas that may include giant cells, a hallmark of the disease, and arterial tertiary follicular organs. Cells of the vessel wall cells, including vascular smooth muscle cells (VSMCs) and endothelial cells, actively contribute to vascular remodeling responsible for vascular stenosis and ischemic complications. This review will discuss new insights into the molecular and cellular pathogenetic mechanisms of GCA, as well as the implications of these findings for the development of new diagnostic biomarkers and targeted drugs that could hopefully replace glucocorticoids (GCs), still the backbone of therapy for this vasculitis.

## 1. Introduction

Giant cell arteritis (GCA) is a granulomatous vasculitis involving medium- and large-caliber arterial vessels [1,2]. GCA typically affects individuals older than 50 years, most frequently women, with a peak incidence between the seventh and eighth decades of life [3]. For a long time, it was thought that GCA was a disease involving only the temporal artery, giving rise to temporal arteritis (TA). Subsequently, it was clarified that GCA can also involve large-caliber vessels. It was therefore underlined that GCA should be classified into a cranial form (c-GCA) and an extracranial or large vessel form (LV-GCA). LV-GCA mainly involves the thoracic aorta and its branches and is the main cause of noninfectious aortitis in humans [4].

Epidemiological data on GCA can sometimes appear controversial because they depend on the different designs of the studies conducted [5]. The prevalence of LV-GCA is probably underestimated because of the difficulty in diagnosing this form of vasculitis compared with TA. However, although there are different data depending on the geographical area considered, through a large meta-analysis, an overall incidence of 10 cases and a prevalence of 51.74 cases per 100,000 people was calculated [6]. The highest incidence is in Scandinavian countries, with a significant north–south gradient in European countries [7]. The higher prevalence of GCA in the female sex was confirmed in a 15-year national retrospective study of inpatients in the United States. Specifically, it found that females accounted for 71.9% of patients with GCA, albeit with a lower mortality risk than males [8].

The onset symptoms of GCA in TA include headache, jaw pain, scalp tension, fever, and weight loss. The most feared complication of c-GCA is transient or permanent blindness due to the involvement of the ophthalmic artery [9,10]. On the other hand, LV-GCA can cause aortic stenosis due to vascular remodeling secondary to inflammation, as well as thrombosis, limb claudication, mesenteric ischemia, and arterial aneurysms [4,11]. All these conditions contribute to a significant increase in mortality.

Although the clinical and epidemiological features of GCA have been partly elucidated, the causes of this vasculitis remain largely unknown. The purpose of this review is to summarize the current evidence on the molecular and cellular mechanisms of GCA, emphasizing both the genetic and epigenetic factors and the immunological and non-immunologic mechanisms responsible for the inflammatory process and vascular remodeling. The implications of this knowledge for the improvement of diagnosis through the identification of potential biomarkers and for the design of effective targeted therapy will also be discussed [12].

## 2. Giant Cell Arteritis and Polymyalgia Rheumatica

A distinctive feature of GCA is its close association with polymyalgia rheumatica (PMR). PMR is a musculoskeletal inflammatory disease characterized by an acute onset with typical signs and symptoms, including morning stiffness of the scapular girdle, anemia, fever, fatigue, and weight loss [13]. Several evidences indicate that the two conditions are not only often associated, but share similar pathogenetic mechanisms [14] and epidemiological characteristics, such as older age of onset, increased incidence and prevalence in northern European countries, and higher prevalence in the female sex [9]. PMR is about 2–3 times more common than GCA and is present in about 50% of patients with this vasculitis and can occur before, at the same time, or after the onset of GCA. On the other hand, GCA is diagnosed in 15–20% of PMR patients [15]. It is also interesting to note that asymptomatic GCA has been demonstrated by biopsy or ultrasonography [16] or detected at the autopsy [17] in a significant number of subjects with PMR, suggesting that the link between the two conditions is probably underestimated.

The hypothesis that the two diseases may be different stages of the same disease was first formulated in 1972 [18]. More recently, based on further experimental evidence, it has been proposed that failure of peripheral tolerance, breakdown of tissue barriers, and granuloma formation constitute successive stages of a single process proceeding from PMR to GCA. This led to the formulation of the combined term GCA-PMR spectrum disease (GPSD) to define a single comprehensive disease entity [19,20]. An international committee recently recommended that the close temporal and pathogenetic correlation between GCA and PMR should be closely considered in a treat-to-target approach [21]. The European Alliance of Rheumatology Associations (EULAR) recently updated its recommendations for the management of GCA in the more general context of large vessel vasculitis [22].

## 3. Genetics and Epigenetics

Early studies on the genetic predisposition to develop GCA focused on the association with the presence of particular HLA alleles. In particular, HLA-DRB1*0401 and HLA-DRB1*0404 alleles were found to be significantly associated with GCA [23]. Subsequently, mutations were identified in non-HLA genes, including gene encoding for inteleukin-10 (IL-10) and vascular endothelial growth factor (VEGF) [24]. In further studies, single nucleotide polymorphisms (SNPs) have been reported, including the variant R620W in the gene encoding for protein tyrosine phosphatase non-receptor type 22 (PTPN22), also known as lymphoid tyrosine phosphatase (LYP) [25]. This protein modulates both T- and B-cell responses. New associations between genetic mutations and GCA have also been described. These include association with those encoding for the NOD-like receptor family pyrin domain containing-1 (NLRP1) and with the genes encoding interluchin-17A (IL-17A) [26,27]. The latter association supports the notion of the critical involvement of IL-17-producing cells, particularly Th17 cells, in the pathogenesis of GCA. In addition, IL-17A is also produced by neutrophils found in the vascular walls of affected patients [28]. In genome-wide association studies (GWAS), genes encoding for plasminogen (PLG) and the alpha 2 subunit of prolyl 4-hydroxylase (P4HA2) have been identified as other genetic factors responsible for susceptibility to GCA [29].

Regarding epigenetic modifications that have been shown to be involved in the onset of GCA, hypomethylation of loci of genes encoding for proteins involved in T-cell receptor (TCR) activation, especially after interaction with the co-stimulatory molecule CD28 and of genes involved in the calcineurin-mediated intracellular pathway, which is critical for the induction of the nuclear transcription factor of activated T cells (NFAT), was observed in the arterial wall of GCA patients [30]. Hypomethylation of a number of genes, including *IFNG*, *TNF,* and *CD40LG*, which are activated by NFAT, was also observed in the arterial wall of affected patients. By histochemical analysis, the transcription factor NFAT1 was found within the nucleus of arterial wall cells in patients with GCA as an expression of its activation. In addition, hypomethylation of genes encoding for interferon gamma, IL-6, IL-21, and the chemokines CCR7 and CCL18, which are a molecular signature of Th1, Th17, macrophage, and dendritic cell activation, strongly support the activation of these cells in GCA [30].

A relatively new line of research is the role of microRNAs (miRNAs) in GCA pathogenesis. In particular, miR-146, miR-155, and miR-21 have been found to be overexpressed in the arterial wall of patients with CGA [31]. It is believed that the role of these miRNAs is to activate pro-inflammatory cells, including T cells, macrophages, and dendritic cells, as well as to participate in vascular remodeling, which also depends on the active intervention of vascular smooth muscle cells (VSMCs) [32,33,34]. Moreover, since a distinguishing feature of GCA is that it affects individuals at an advanced age, age itself must be considered a key epigenetic factor in the pathogenesis of this vasculitis. As is well known, the immune system tends to change its efficiency with age, undergoing a regressive phenomenon called immunosenescence. In addition to the reduced ability to respond to external stimuli such as pathogens, aging of the immune system also corresponds to so-called inflammaging, i.e., the reduced control of inflammatory responses due to the decreased efficiency of the mechanisms that control this response [35,36]. In this regard, both T-cell DNA hypomethylation and increased miRNA expression are phenomena associated with the aging of the immune system [31,37].

Another epigenetic factor reported to be possibly involved in the genesis of GCA is the microbiota. The idea that GCA might be induced by infectious factors was first hypothesized following the observation of the seasonality of the disease [38]. In recent years, it has emerged that the close relationship between gut dysbiosis and immune response could participate in the pathogenesis of many autoimmune diseases [39]. A microbiota has also been shown to be physiologically present in large vessels previously considered sterile [40]. It is possible that under particular conditions, the vascular microbiota, interacting directly with vascular immune cells, may promote the onset of GCA [40]. In a very recent study, specific alterations in the blood microbiome were described that differed between GCA and healthy subjects or those with Takayasu’s arteritis, further emphasizing the possible role of the microbiota in the pathogenesis of large vessel vasculitis (LVV) [41]. The first pathogen thought to play a crucial role in the pathogenesis of GCA was the varicella-zoster virus (VZV) [42]. Although VZV proteins have been found within the vascular wall in patients with GCA [43,44,45], this result was not confirmed by all studies [46,47]. Another clue suggesting the involvement of microorganism-derived antigens is the reported observation of increased incidence of GCA after influenza vaccination [48] and, more recently, COVID-19 vaccination [49].

## 4. Immunopathogenesis

### 4.1. The Role of Innate Immune System

#### 4.1.1. The Macrophages and Neutrophiles

Macrophages have been found to infiltrate the arterial wall of patients with GCA. These cells of the innate immune system are one of the main cell types involved in granuloma formation and are recruited into the vessel wall by DCs and T cells [34,50]. Attracted to the site of inflammation by soluble factors, particularly chemokines, macrophages are, in turn, capable of producing high levels of active substances, including matrix metalloproteinase 9 (MMP9) [51]. This enzyme degrades the arterial wall matrix, allowing the macrophages themselves to penetrate deeper into the adventitious layer of the vessel, where they can induce local recruitment of additional proinflammatory cells. M1-type macrophages, characterized by high proinflammatory activity, tend to localize between the media and adventitia, amplifying damage to the vascular wall through the production of cytokines IL-1 and IL-6 and of reactive oxygen species (ROS). Anti-inflammatory M2-type macrophages, on the other hand, tend to localize at the border between the media and vascular intima [50,52,53]. This subgroup of macrophages produces high amounts of VEGF. Endothelial changes induced by VEGF release are thought to be crucially involved in the vascular remodeling observed during GCA, with thickening of the vascular wall [50,54]. It should be noted that in GCA, macrophages tend to form special multinucleated cells after fusion of activated cells, so-called giant cells, which are the hallmark of this vasculitis. These cells usually result from resistance to degradation of the phagocytosed material [55]. Giant cells have been found in about one-third of temporal artery biopsies from patients with c-GCA [56,57,58].

Neutrophils also play an important role in the pathogenesis of GCA. Neutrophil activation after stimulation by danger signals results in the local release of pro-inflammatory cytokines, including IL-6 [59] and IL-17A [28,60]. Neutrophils also produce enzymes capable of disrupting the extracellular matrix and attacking invading pathogens. Activation of nicotinamide adenine dinucleotide phosphate (NADPH) oxidase produces a respiratory burst through the production of highly reactive oxygen species (ROS) necessary to kill microbes [28,61,62,63]. Neutrophils also form the neutrophil extracellular traps (NETs). NETs are composed of filaments of nuclear material capable of trapping and killing various pathogens. The formation of NETs (NETosis) can follow the apoptosis of neutrophils or occur together with the preservation of neutrophil viability. NET formation can be initiated by various stimuli, including microorganism components, cytokines, immune complexes, autoantibodies or platelets. Exposure to such signals induces disassembly of the neutrophil cytoskeleton, chromatin decondensation with citrullination of histones, and formation of bactericidal substances such as myeloperoxidase (MPO) and pro-coagulant factors as well. The process of NETosis depends strictly on ROS production by NAPDH oxidase and/or activation of mitochondria [64]. NET formation may be an important source of autoantigen, as demonstrated in ANCA-associated vasculitis [65]. Recently, NETs containing pro-inflammatory cytokines were found in the temporal biopsy of GCA patients [28].

#### 4.1.2. The Dendritic Cells

A crucial role in the pathogenesis of GCA is played by dendritic cells (DCs) that reside in the space between the media and adventitia of the arterial wall [53,66]. Dendritic cells express toll-like receptors (TLRs) on their surface, which can recognize specific pathogen-associated molecular patterns (PAMPs) derived from components of microorganisms and/or damage-associated molecular patterns (DAMPs) that recognize various proteins, including fragments of necrotic self-cells [67,68]. Large numbers of DCs have been shown to reside at the level of the vascular wall in patients with GCA [69]. It has been shown that such cells can be recruited to this site through the expression of the C-C chemokine receptor 7 (CCR7). This receptor selectively binds its ligands CCL19 and CCL21, which are abundantly present at the level of vascular tissue. CCL2 is also involved in the recruitment of several cells of the innate and adaptative immune system to the artery wall [70,71]. Once activated, DCs can induce the activation of macrophages with subsequent amplification of the inflammatory cascade [66,72,73]. In addition, DCs could play a key role in the presentation of a putative antigen to T lymphocytes, thus contributing to their activation of cytokine production [74] and granuloma formation [75].

It has also been suggested that DCs are abnormally activated in patients with GCA due to a deficit in the expression of programmed death ligand 1 (PD-L1), which normally promotes immunosuppression [76,77]. In this context, PD-L1-deficient myeloid DCs were hypothesized to facilitate the recruitment of CD4+ T cells into the vascular wall in GCA [74] while plasmacytoid DCs could activate CD8+ T cells [78]. In support of the role of defective PD-L1 expression by DCs in the pathogenesis of GCA, the possible occurrence of this vasculitis was observed during cancer immunotherapy with immune checkpoint inhibitors (ICIs). These agents selectively block the interaction between PD-L1 and the PD-L1 ligand. In a large review study, an incidence of GCA/PMR of 0.26% of total cases of immunotherapy-related adverse events (Ir-AEs) associated with ICI treatment was reported [79,80,81]. In a chimeric animal model of severe combined immunodeficiency (SCID) transplanted with human arteries from patients with GCA, the blocking of PD-1 enhanced vascular inflammation and cytokine production, further supporting the role of the lack of PD-L1/PD-1 interaction in GCA [82].

### 4.2. The Involvement of Adaptive Immunity

#### 4.2.1. The Role of T Cells

The first clue suggesting the involvement of T cells in GCA was the finding that the immunohistology lesions in the wall of vessels affected by this vasculitis are granulomatous in nature [1]. Granulomas are known to consist of both cells of the innate system, such as macrophages, and cells of the adaptive system, including T cells that aggregate to form well-ordered structures in the course of inflammation. Further characterization of the subtypes of T cells infiltrating the arterial wall revealed the presence of two major subsets of CD4+ T cells, namely Th1 and Th17 cells [75]. Although granulomas are generally detected in the course of infections [83], an infectious response to a viral/bacterial antigen has never been convincingly demonstrated in the case of GCA, as discussed above in more detail [84]. However, early research aimed at characterizing the antigenic specificity of T cells in GCA found that it was restricted to a few antigens [85]. Supporting the role of a microbial antigen in shaping the Th1 cell repertoire in GCA, at least as an initial disease trigger, is the finding that the TCR rearrangement of T cells infiltrating the right and left temporal artery was identical in arteries on both sides of the skull in individual patients [86].

Th1 and Th17 cells are believed to play different roles in the pathogenesis of GCA. Th1 cells are characterized by the production of IFN-γ, and their differentiation is controlled by IL12/IFN-γ axis [87,88]. It has been reported that the presence of high levels of IFN-γ in the vascular wall of patients with GCA is strongly associated with the tendency to develop ischemic vascular injury [89]. In addition, this cytokine has been found to be elevated years before the onset of GCA [90]. It should be noted that, unlike Th17 cells, Th1 cells are not sensitive to steroid therapy. This fact confirms an additional unmet need for GCA therapy and justifies the effort to identify new therapeutic agents that can also be effective on this important subgroup of cells [53,91].

As for Th17 cells, they have been found in high concentrations in both the peripheral blood and the inflammatory infiltrate of the vascular wall of patients with GCA [75,92,93]. Th17 cells are characterized by the production of IL-17 [94], an interleukin that facilitates macrophage recruitment and defense against infections [95,96]. IL-17 induces the production by target cells of many pro-inflammatory molecules, such as IL-6 and chemokines, including CXCL8 [97], thereby amplifying the downstream inflammatory reaction. It has been clearly established that Th17 cells are involved not only in defense against pathogens but also in the induction of immunopathological phenomena and autoimmunity [98,99]. The characteristic plasticity of Th17 cells in transforming, under particular stimuli from the microenvironment, into cells with different phenotypes, such as Th1 cells or regulatory T cells (Treg) [100] makes it further complex to study their role in the immunopathogenesis of GCA and autoimmunity in general.

An important role in the pathogenesis of GCA is played by Treg. As mentioned above, these cells are known to be able to differentiate into Th17 cells and vice versa [101]. It was clarified that IL-6, together with tissue growth factor-β, promotes polarization toward the Th17 cell phenotype, whereas IL-6 alone induces differentiation of Treg [102]. The presence of IL-23 in the microenvironment produced by cells of the innate system further inhibits the expression of the transcription factor forkhead box P3 (FOXP3) necessary for Treg differentiation. It facilitates the maintenance of the Th17 phenotype and its survival [103,104]. It has been shown that Treg present at the level of the vessel wall in GCA are unable to perform their regulatory function. Therefore, these cells participate in the pathogenic process instead of inhibiting the pro-inflammatory activity of T-cell subpopulations. Interestingly, therapeutic blockade of IL-6R is able to inhibit these pathogenic Treg in CGA, restoring their anti-inflammatory function, whereas this is not the case with GC therapy [105].

It should be further emphasized that the differentiation of T cell precursors into the various T cell subpopulations depends critically on soluble factors produced by cells of the innate immune system, and in particular, DCs [106]. DCs can thus be considered the conductors of the T-cell response orchestra. As already discussed, the differentiation of Th0 cells into Th1 cells depends mainly on the presence of IL-12, while the differentiation and stabilization of Th17 cells depend on the presence of IL-23 [107]. It has been hypothesized that in GCA, as yet unknown stimuli capable of activating different TLRs on DCs may determine the location and severity of this vasculitis through the induction of responses by specific subpopulations of T cells [66,69,108]. In this regard, studies in animal models have shown that TLR4 stimulation on dendritic cells induces transmural arterial inflammation, whereas TLR5 activation is rather related to perivascular inflammation [72].

Other subpopulations of CD4+ T cells that have been described to play an important role in GCA are Th2 and Th9 cells, which produce IL-33 and IL-9, respectively [109,110]. Further studies are needed to clarify the role of these CD4+ T-cell subsets in the pathogenesis of GCA.

CD8+ T cells have been less studied than CD4+ T cells in terms of their pathogenetic role in GCA. Although it has been reported that these cells may not play a significant role in the inflammatory process of GCA due to their anergy related to the advanced age of GCA patients [111,112,113], on the contrary, many studies support the central role of both cytotoxic and regulatory CD8+ T cells in the pathogenesis of this vasculitis. CD8+ T cells with increased ability to produce granzyme A have been reported to be present in the peripheral blood of patients with GCA, and their numbers decreased after GC therapy [114]. Temporal artery tissue has also been shown to express natural killer group 2D (NKG2D) ligand that can stimulate the cytotoxic activity of NKG2D receptor-expressing CD8+ T cells [115]. As for CD8+ regulatory cells (CD8+ Treg), these are dysfunctional in GCA, like their CD4+ counterpart [116]. The reduced anti-inflammatory activity of this anti-inflammatory CD8+ T cell subset has been attributed to NADPH oxidase 2 deficiency [117].

Finally, it should be noted that memory T cells residing in the vascular wall of patients with GCA have been shown to contribute to the renewal and persistent presence of proinflammatory T cells at the vascular inflammatory site [118,119,120].

#### 4.2.2. The Role of B Cells

Initial studies had suggested a possible role of autoantibodies produced by B cells in the pathogenesis of GCA. These findings, however, have not subsequently been confirmed [121,122]. In more recent studies, B cells and plasma cells have been identified in high numbers in the temporal artery of patients with this form of GCA [123]. In this case, these cells were found to form together with T lymphocytes, dendritic cells, and high endothelial venules (HEVs). These follicle-like structures have been termed tertiary arterial lymphoid organs (ATLOs) [124]. ATLOs are predominantly located near granulomas, suggesting that communication between the two structures can occur. Such proximity may, therefore, facilitate macrophage activation in granulomas through the arrival of cytokines and other soluble factors produced by cellular interactions within ATLOs.

In further support of the key role of B cells in the pathogenesis of GCA, the B-cell-activating growth factor (BAFF) and the proliferation-induced ligand APRIL, both produced by endothelial cells and VSMCs, have been found in high concentrations at the tissue level in patients with GCA [124]. In a recent study, it was shown that, in the active phase of the disease, high numbers of B cells can be identified in the arterial wall of large vessels of GCA patients while decreasing in peripheral blood, demonstrating tissue-selective recruitment of these lymphocytes [125,126]. The potential role of B cells in the pathogenesis of GCA is also suggested by the description of two cases that significantly improved following therapy with the anti-CD20 antibody rituximab that depletes B cells [127,128]. The absence of specific antibodies during GCA suggests that these cells play a different role in the immune response rather than contributing to humoral immunity, including the production of soluble factors or antigen presentation to T lymphocytes [129,130,131]. The role of B cells in the pathogenesis of GCA is further underscored by the observation that during the acute stages of the disease, their numbers in the peripheral blood tend to decrease. This denotes their redistribution to the vascular tissue site of inflammation [125,132]. Figure 1 summarizes the cellular mechanisms involved in the pathogenesis of GCA.

## 5. The Role of Vascular Arterial Wall Components

A significant role in the pathogenesis of GCA is played by the cells of the arterial wall itself. In this regard, it has been shown that vascular stenosis, which is the main complication of GCA, is caused by remodeling of the vessel wall. The tunica media is progressively destroyed, while the intima undergoes thickening due to myofibroblast proliferation and protein deposition in the extracellular matrix, leading to vessel occlusion [3,133]. A key role of macrophages has been identified in this process. These cells, activated by granulocyte-macrophage colony-stimulating factor (GM-CSF), in turn activate VSMCs. VSMCs produce matrix metalloproteinase (MMP)-2 and MMP-9, which contribute to the destruction of the media and internal elastic lamina [134]. In addition, IFN-γ secreted by Th1 cells activates VSMCs to produce other important factors involved in vascular remodeling. These include platelet-derived growth factor (PDGF), VEGF, and endothelin-1 [135]. PDGF induces the proliferation of VSMCs and their migration into the intimal layer, VEGF promotes angiogenesis, and endothelin-1 promotes the differentiation of VSMCs into myofibroblasts. This complex mechanism eventually leads to intimal hyperplasia and subsequent vessel occlusion. The role of many of these factors in vascular wall remodeling has been indirectly demonstrated in experimental models using PDGF or endothelin-1 inhibitors that resulted in the blockade of VSMC migration and proliferation [136,137]. An important role could also be played by perivascular mesenchymal cells. Indeed, dysregulation of these anti-inflammatory cells would facilitate inflammation and fibrosis, thus contributing to vascular damage and remodeling in GCA [138]. All this evidence suggests that the vascular component, in addition to the immunologic component, may offer attractive targets for innovative therapies of GCA.

## 6. Implications of Pathogenetic Findings in Clinical Practice

The new knowledge about the pathogenesis of GCA is not only of interest to bench scientists but also has an important impact on the management of this vasculitis at the bedside. Regarding c-GCA, although imaging techniques such as ultrasound, magnetic resonance imaging (MRI), and computed tomography (CT) can provide significant help, the diagnostic gold standard is still the arterial biopsy. However, it has been shown that interpretation of the histologic finding can sometimes lead to erroneous conclusions because of the difficulty in differential diagnosis with GCA-like conditions affecting the temporal artery, including antineutrophil cytoplasm-associated vasculitis [65,139]. Even more complex is the diagnosis of LV-GCA, which relies solely on imaging techniques, although in this context, ^18^fluoro-2-deoxy-D-glucose (FDG)-PET/CT has recently demonstrated excellent accuracy [140]. However, the availability of blood tests that can identify specific biomarkers of GCA represents an unmet clinical need, as such tests could allow rapid and specific diagnosis, less invasive than a biopsy, and feasible in those centers that lack facilities for advanced imaging investigations. As highlighted by the present review, the pathogenetic mechanisms of GCA emphasize the importance of high-grade inflammation as the final element of vascular damage. For this reason, classical inflammatory indices such as C-reactive protein (CRP) and erythrocyte sedimentation rate (ESR) are considered key markers of the disease to date. However, these biomarkers have been challenged by some studies that have diagnosed the presence of GCA, even in the absence of these classical inflammatory markers [141].

Given the role of B lymphocytes in the pathogenesis of GCA, autoantibodies that could facilitate diagnosis were also sought. Among these, anti-cardiolipin and anti-ferritin antibodies were proposed as potential biomarkers, although their presence proved to be nonspecific [142].

In a large study, two inflammation-related substances were identified as having greater diagnostic specificity. These are the aforementioned IFN-γ and the monocyte chemoattractant protein 3 (MCP-3). These potential biomarkers have been found to be elevated in the serum of patients even years before disease onset [90]. Because of the important role played by macrophages in the pathogenesis of GCA, these markers are potential candidates for serologic diagnosis of this vasculitis.

Several other soluble factors have been identified at high concentrations in the serum of patients with GCA, including IL-1β, IL-6, IL-23, VEGF, and chitinase-3- like protein or YKL-40. A very comprehensive review of biomarkers in GCA has been recently published [143]. The increased synthesis of these substances depends on the activation of pro-inflammatory cells involved in the pathogenetic mechanisms of GCA. Of particular interest is the finding that interleukin-6 (IL-6) and interleukin-23 (IL-23) have been found at very high levels in the serum of GCA patients [144]. The levels of IL-6 were found to be correlated with the level of inflammation in GCA [145]. Based on these and other findings, the anti-IL-6 receptor (IL-6R) antibody tocilizumab was tested for the treatment of GCA. To date, tocilizumab is the only biologic agent approved by regulatory agencies for the treatment of GCA due to the results obtained in the Actemra Giant Cell Arteritis Study (GIACTA) [146,147]. In a secondary analysis of data obtained during this study, therapy with tocilizumab was also able to restore the impaired anti-inflammatory function of Treg [105]. The primary endpoint, that is, a significantly higher percentage of patients in GC-free remission at 52 weeks compared with the placebo group, was achieved in the GIACTA trial. However, long-term data are scarce, as are effects on complications, such as thoracic aneurysm formation. In addition, disease recurrence occurred in about 30% of treated patients, and 5–10% of treated subjects had to discontinue treatment due to adverse effects. Therefore, despite the positive results obtained, alternative treatments need to be found. Sarilumab, another anti-IL6R monoclonal antibody, recently shown to be effective in a phase 3 study and thus approved by the FDA for the treatment of PMR [148] is also under investigation for GCA treatment (ClinicalTrials ID: NCT03600805).

Also of interest is the increase in the level of IL-23, found elevated in the serum of patients with GCA [149]. Since this cytokine secreted by dendritic cells is necessary for the differentiation and survival of Th17 cells, which exert their pathogenic role through IL-17 synthesis, this suggested the possible use of biologics targeting these interleukins in therapy. Randomized controlled trials have been conducted on the use of ustekinumab, a human monoclonal antibody capable of blocking both IL-12, which is required for Th1 cell development, and Il-23, which is required for maintenance of the Th17 cell phenotype. This biologic drug thus appears to be an ideal candidate for the treatment of GCA targeting both T cell subpopulations. Studies to date have shown positive results with discontinuation of steroid therapy in patients during treatment [150]. However, it is still premature to draw definitive conclusions on the use of this biologic for the treatment of GCA as the positive results obtained may have been influenced by the concomitant intake of GC by the treated patients. Guselkumab, another anti-IL23 biologic, is currently under study (ClinicalTrial.gov ID: NCT0463447) [151].

As described previously, IL-17A is a key cytokine in the pathogenesis of GCA. This interleukin is produced by different cell types of innate and adaptive immunity, as well as endothelial cells and VSMCs. A randomized controlled trial (RCT) is underway to evaluate the efficacy and safety of secukinumab, an anti-IL17A monoclonal antibody, with encouraging preliminary results yet to be published (ClinicalTrials ID: NCT04930094).

Clinical studies have also been conducted on fusion protein abatacept, which interferes with the interaction of CD80/CD86 with co-stimulatory receptor CD28, which is necessary for T-cell activation. Blocking co-stimulation of T cells by dendritic cells might be a good strategy to inhibit autoreactive T cells, which are probably implicated in the pathogenesis of GCA. In one randomized controlled trial (RCT), abatacept showed moderate activity in improving the arterial inflammatory status of patients with GCA [152]. Regarding the role played by type-I interferon in GCA, particularly IFN-α, the interferon signature, that is, the activation of those genes specifically induced by the action of this cytokine at the level of aortic tissue, was studied. The type-I IFN signature was found in the inflamed tissue of biopsies from patients with GCA but not in unaffected subjects [153]. However, there are still no studies available on the possible use of therapies targeted to type-I IFN in GCA.

Of considerable interest is the role of the Janus kinase/signal transducer and activator of the transcription (JAK/STAT) pathway in the pathogenesis of GCA. Indeed, the JAK/STAT signaling pathway appears to be up-regulated in patients with this vasculitis [153]. Because this transduction pathway leads to nuclear transcription of genes encoding multiple cytokines involved in the inflammatory process, its inhibition appears particularly interesting. The recent availability of selective inhibitors of the JAK/STAT signaling pathway, which have so far been approved for the treatment of inflammatory rheumatic diseases such as rheumatoid arthritis, psoriatic arthritis, and ankylosing spondylitis, has prompted a push to study their efficacy in GCA as well. Tofacitinib, a JAK inhibitor (JAKi) that predominantly inhibits JAK1 and JAK3 molecules has been shown to ameliorate aortitis due to GCA by inhibiting the action of T lymphocytes [119]. In another study, baricitinib, an inhibitor with selectivity for JAK1/JAK2, currently approved for the treatment of rheumatoid arthritis, was seen to allow treated patients to significantly reduce steroid use. [154]. Another study (SELECT-GCA) is underway to determine the role of upadacitinib, a selective JAK1 inhibitor, in the treatment of this vasculitis (ClinicalTrials ID: NCT03725202). Table 1 shows the targeted drugs currently approved or under study for the treatment of GCA.

Of particular interest are therapies aimed at inhibiting vascular remodeling. Since GM-CSF appears to be crucial in the destruction of media by macrophages and is involved in the genesis of intimal hyperplasia and neovascularization, its blockade with the specific monoclonal antibody mavrilimumab seems to be an attractive therapeutic option. A phase II RCT has been recently completed with encouraging results (ClinicalTrials ID: RCT NCT). Instead, studies are underway on bosentan, an endothelin-1 receptor antagonist. This substance produced by vascular endothelial cells is a potent vasoconstrictor and is involved in the vascular remodeling of GCA. The current phase III (ClinicalTrials ID: RCT NCT03841734) aims to determine whether this substance can prevent blindness in subjects with TA.

Finally, it can be speculated that additional therapies may be available in the future that take advantage of new insights into the pathogenesis of GCA. These could include the administration of exogenous IL-2 to reverse Treg dysfunctionality, as pioneered in other rheumatic diseases [155], or the use of PD-1-stimulating antibodies to restore the immune checkpoint. One such antibody, peresolimab, is currently being evaluated in rheumatoid arthritis [156].

## 7. Conclusions

Although GCA is the most frequent vasculitis of the great vessels, the etiology is still unknown, and the pathogenesis has not yet been fully elucidated. Several predisposing genetic factors have been identified, but it has emerged that epigenetic factors are essential in triggering the onset of the disease. The advanced age of GCA patients and the close association with PMR are other aspects that require further study to be adequately explained, as well as why the disease may have a different arterial localization in different patients. However, new knowledge, albeit partial, has led to the approval of innovative targeted therapy, such as, in particular, the use of the anti-IL6R monoclonal biologic agent tocilizumab. Anti-IL-17 antibodies are at an advanced stage of study, and great expectations are placed on JAK inhibitors. Further studies are needed to draw an increasingly accurate picture of the pathogenesis of GCAs. Such data will be needed to identify new diagnostic biomarkers, improving the diagnostic accuracy of GCAs, and to set up increasingly effective therapies that may allow a significant percentage of patients not to depend on the continued use of GCs.

## Figures and Tables

**Figure 1 cells-13-00267-f001:**
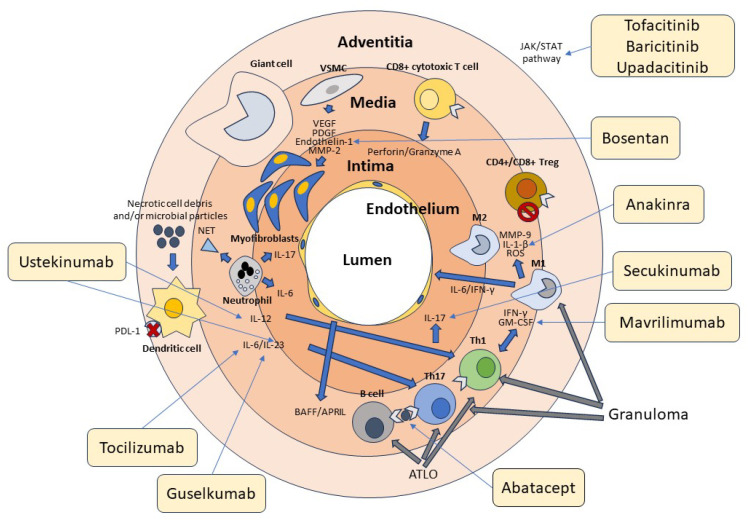
Cellular and molecular pathogenesis of GCA. Cells of the innate and adaptive immune system and the vascular wall contribute to the pathogenesis of GCA. Adventitious dendritic cells are activated by unknown molecules of microbial and/or cellular origin. Dendritic cells in GCA are typically defective in the expression of the immunosuppressive surface molecule PD-L1, as indicated in the figure by the red cross. Dendritic cells produce IL-12, which promotes the differentiation of Th1 cells, and Il-6 and IL-23, which contribute to the differentiation and stabilization of the phenotype of Th17 cells. T lymphocytes are activated by both dendritic cells and B lymphocytes through the presentation of a putative antigen. Th1 cells produce IFN-γ and GM-CSF, while Th17 cells produce IL-17. These cytokines activate M1 and M1 macrophages, which in turn produce MMP-9, IL1-β, and ROS, contributing to media destruction. Some macrophages, unable to kill the phagocytosed material, transform into giant cells. CD4+ and CD8+ Treg cells participate in the inflammatory reaction, being deficient in their immunosuppressive function, as indicated by the stop symbol. CD8+ cytotoxic T cells and neutrophils producing proinflammatory cytokines and NETs play an additional role in the pathogenesis of GCA. At the vascular level, damaged VSMCs produce VEGF, PDGF, endothelin-1, and MMP-2, which promote their differentiation into myofibroblasts. These cells cause thickening of the intima and subsequent vascular stenosis. Many cytokines recognize JAK-associated receptors and amplify the inflammatory cascade through the activation of the JAK/STAT signaling pathway. T and B cells aggregate to form tertiary follicular structures (ATLO), whereas T cells and macrophages are the main components of granulomas. The light-yellow boxes show the main drugs available to date that inhibit the various factors involved in the pathogenesis of GCA, as indicated by the thin arrows. Of these, only tocilizumab has been officially approved for the treatment of this vasculitis.

**Table 1 cells-13-00267-t001:** Targeted drugs approved or in clinical trials for the treatment of GCA.

Drug	Target	Structure	Approved	Main Reference
Tocilizumab	IL-6-R	MOAB	Yes	[146]
Sarilumab	IL-6-R	MOAB	No	NCT03600805
Secukinumab	IL-17A	MOAB	No	NCT04930094
Ustekinumab	IL1-12/IL-23	MOAB	No	[150]
Guselkumab	IL-23	MOAB	No	NCT04633447
Mavrilimumab	GM-CSF	MOAB	No	NCT03827018
Abatacept	CD80/CD86	FP	No	[152]
Anakinra	IL-1β	RRA	No	NCT02902731
Tofacitinib	JAK1/JAK2/JAK3	SMOL	No	[119]
Baricitinib	JAK1/JAK2	SMOL	No	[154]
Upadacitinib	JAK1	SMOL	No	NCT03725202
Bosentan	Endothelin-1	RRA	No	NCT03841734

IL = interleukin; R = receptor; GM-SSF = granulocyte-macrophage colony-stimulating factor; JAK = Janus kinase; MOAB = monoclonal antibody; RRA = recombinant receptor antagonist; FP = fusion protein; SMOL = small molecule.

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
