# Peer review of "Giant Cell Arteritis: Advances in Understanding Pathogenesis and Implications for Clinical Practice"

_cells, 2024, doi:10.3390/cells13030267_

Round 1
Reviewer 1 Report
Comments and Suggestions for Authors
This review paper is very well written and deeply informative for the Giant cell arteritis. The relation to PMR is well cited, but n to further explored. Furthermore, a reaction to the microbiome is cited, but details are not named. In summary, it is a very good review f current knowledge in the field.
1. as a review it is not considered as a report of original research, but it is a review of existing evidence of the pathogenetic mechanisms of GCA.
2. As compared to current literature includes more recent genetics and epigenetics, single nucleotide polymorphisms that activate T cells, as well as NOD-like receptor family pyrin domain containing 1 and genes encoding IL17 A. Epigenetic modifications leading to activation of T cells NFAT, IFNg, TNF, CD40-L etc. by hypomethylation and consecutively micro RNAs are well described. The role of regulatory T cells is briefly touched, since treatment with ICI may induce GCA and depletion of regulatory T cells in an animal model increases the pathology of the transferred vascular piece.
3. The review includes new data on genetics, natural and adaptive immunity including a role of B cells and consequently a list of new studies of treatment with antibodies and small molecules.
4. The role of infections is less well confirmed: varicella-zoster and “microbiota” without further specification, but also as a consequence of vaccination. The formation of giant cells and granulomas is often the result of resistance to degradation of phagocytosed material. The role of neutrophils and their extracellular traps needs to be defined. Finally the perivascular mesenchymal stromal cells are not displaced that may have an immunomodulatory role.
The figure is not informative and does not add essential facts. It should give relevant information on the pathogenetic process and the potential interactions with new treatments.
5. In summary the review gives an interesting overview on genetics and immunology without an explanation of the relevant mechanisms.
Comments on the Quality of English LanguageThe quality of English is perfect for non-English speaking reviewer. There are a few spelling mistakes. The toll like receptors are falsely named tool like receptors.
Author Response
REVIEWER#1
(changes to the text have been highlighted in green)
Query: The formation of giant cells and granulomas is often the result of resistance to degradation of phagocytosed material. Answer: As suggested, a commentary on the origin of mononuclear giant cells and a related literature reference has been added.
Query: The role of neutrophils and their extracellular traps needs to be defined. Answer: The discussion of neutrophil function and NET formation has been expanded as suggested, and the most recent and relevant literature references have been added.
Query: The perivascular mesenchymal stromal cells are not displaced that may have an immunomodulatory role. Answer: A commentary on the role of perivascular mesenchymal cells on inflammation and vascular remodeling in GCA, with references, has been included in the new review chapter entitled "5. Role of Arterial Vascular Wall Components."
Query: The figure is not informative and does not add essential facts. It should give relevant information on the pathogenetic process and the potential interactions with new treatments. Answer: The figure has been modified while maintaining its schematic style, enriching it with additional pathogenetic mechanisms and clarifying those already illustrated. Therefore, the figure legend has been completely rewritten.
Query: The toll like receptors is falsely named tool like receptors. Answer: The typo has been corrected
Reviewer 2 Report
Comments and Suggestions for Authors
See document

Author Response
RESPONSE TO REVIEWER#2
(Changes to the the text have been highlighted in yellow)
Query: I would appreciate if the authors can mention if there are new data brought by their review, based on the latest scientific advances and if/ how can they contribute to the existing body of knowledge in the field. Answer: An introductory sentence has been added and, later in the text, a specific discussion of the emerging role of vascular smooth muscle cells and endothelial cells in promoting vascular remodeling through the synthesis of various soluble factors, as reported in the most recent scientific literature.
Query: The authors pay full attention to innate and adaptative immunity, immune cell populations and their activation patterns, driving the pathogenesis of the disease. However, the picture is not complete, as long as we are speaking about a systemic vasculopathy-blood vessels have their own reactivity, through their specific components like endothelial cells, fibroblasts, vascular smooth muscle cells, etc. I suggest authors to include data on the role of vascular arterial wall components, because of their role in overall assessment, highlighting advancements in understanding the vascular changes associated with GCA. Answer: A chapter on the role of arterial vascular components in the pathogenesis of GCA, has been added to the text, with relevant literature references.
Query: I also recommend a clear and more detailed information on the therapeutic strategy, targeting each of the 3 entities (i2 immune and vascular) involved in GCA pathogenesis. Answer: The chapter on innovative therapeutic strategies for GCA has been revised and expanded to include vascular therapeutic strategies (anti-GM-CSF, anti-Endothelin-1). Table 1, which summarizes ongoing studies, has been updated accordingly.
Query: I suggest authors to carefully revise all throughout the text. (ex-line 58/58- characterized is used twice, replace one of them) Answer: The entire text has been double-checked. The repetition of line 58 has been removed.
Reviewer 3 Report
Comments and Suggestions for Authors
This is an interesting article by Marino Paroli et al.
I would like to address a small number of suggestions to you.
General recommendations
Please correct all references according to journal stile.
References should be described as follows:
Journal Articles:
1. Author 1; Author 2. Title of the article. Abbreviated Journal Name Year; Volume: page range.
Abstract
Page 2 line 94
When we describe IL- 17-producing cells, except to Th17 cells, we will also report neutrophils. Please analyze the article by Palamidas et al.
Page 3 lines 128-129
The role of microbiota as another epigenetic factor of disease is very interesting. Please analyze in depth the role of microbiome in GCA before you start to write about the pathobiota and varicella-zoster virus (VZV). An interesting article in this field was written by Anne Claire Desbois at al. (doi: 10.1038/s41598-021-84725-5.).
Page 4, lines 161-164
Neutrophils play a critical role in the pathogenesis of GCA. The description of their role in only 4 lines (page 4, lines 161-164) does not reflect to significant role that play neutrophils. Please analyzed the role of Neutrophils as a sores of interleukins IL-6, IL-17A.
Please rewrite this paragraph with more data.
Page 7
Figure 1 is too simplified for journal with IF 6. Please draw new figure.
Page 8, line 354 and line 361
Is it necessary to write ID numbers for some clinical trials with too large font ?
Page 9, line395
Table 1
Secukinumab is monoclonal anti IL17A, not IL-7A. Please correct.
Author Response
REWIEVER #3
(Changes to the text have been highlighted in light blue)
Query: Please correct all references according to journal stile. References should be described as follows: Journal Articles: Author 1; Author 2. Title of the article. Abbreviated Journal Name Year; Volume: page range. Answer: The style of the bibliography has been formatted according to the editorial standards of the journal Cells.
Query: Page 2 line 94 When we describe IL- 17-producing cells, except to Th17 cells, we will also report neutrophils. Please analyze the article by Palamidas et al. Answer: The production of IL-17A by neutrophils has been added both in the indicated paragraph and later in the text, and the suggested bibliographic entry has been included in the references.
Query: Page 3 lines 128-129
The role of microbiota as another epigenetic factor of disease is very interesting. Please analyze in depth the role of microbiome in GCA before you start to write about the pathobiota and varicella-zoster virus (VZV). An interesting article in this field was written by Anne Claire Desbois at al. (doi: 10.1038/s41598-021-84725-5.). Answer: The role of the microbiome in the pathogenesis of GCA was addressed in more detail and discussed before VZV infection, as indicated by the reviewer. Some literature references have been added, including the suggested one.
Query: Page 4, lines 161-164
Neutrophils play a critical role in the pathogenesis of GCA. The description of their role in only 4 lines (page 4, lines 161-164) does not reflect to significant role that play neutrophils. Please analyzed the role of Neutrophils as a sores of interleukins IL-6, IL-17A. Please rewrite this paragraph with more data. Answer: The text regarding the role of neutrophils has been expanded extensively (it is highlighted in green because this comment was also raised by another reviewer). The production of IL-6 and Il-17A by neutrophils has been added and highlighted in light blue with relevant literature references.
Query: Page 7 Figure 1 is too simplified for journal with IF 6. Please draw new figure. Answer: Figure 1 has been enriched with additional pathogenetic mechanisms, and those already illustrated have been clarified. The stylized design has been retained to make the content maximally understandable. The legend has been completely rewritten according to the new contents of the figure.
Query: Page 8, line 354 and line 361 Is it necessary to write ID numbers for some clinical trials with too large font? Answer: The wide characters were an artifact that appeared after the manuscript was sent to the journal. The font size was therefore corrected and made consistent with that of the rest of the text.
Query: Page 9, line395 Table 1 Secukinumab is monoclonal anti IL17A, not IL-7A. Please correct. Answer: The typo has been corrected in Table 1.
Round 2
Reviewer 2 Report
Comments and Suggestions for Authors
See the attached file.
